# Time Effects of Global Change on Forest Productivity in China from 2001 to 2017

**DOI:** 10.3390/plants12061404

**Published:** 2023-03-22

**Authors:** Jiangfeng Wang, Yanhong Li, Jie Gao

**Affiliations:** 1College of Life Sciences, Xinjiang Normal University, Urumqi 830054, China; 2Key Laboratory of Earth Surface Processes of Ministry of Education, College of Urban and Environmental Sciences, Peking University, Beijing 100871, China

**Keywords:** global warming, greenhouse effect, productivity, spatialtemporal change

## Abstract

With global warming, the concentrations of fine particulate matter (PM_2.5_) and greenhouse gases, such as CO_2_, are increasing. However, it is still unknown whether these increases will affect vegetation productivity. Exploring the impacts of global warming on net primary productivity (NPP) will help us understand how ecosystem function responds to climate change in China. Using the Carnegie-Ames-Stanford Approach (CASA) ecosystem model based on remote sensing, we investigated the spatiotemporal changes in NPP across 1137 sites in China from 2001 to 2017. Our results revealed that: (1) Mean Annual Temperature (MAT) and Mean Annual Precipitation (MAP) were significantly positively correlated with NPP (*p* < 0.01), while PM_2.5_ concentration and CO_2_ emissions were significantly negatively correlated with NPP (*p* < 0.01). (2) The positive correlation between temperature, rainfall and NPP gradually weakened over time, while the negative correlation between PM_2.5_ concentration, CO_2_ emissions and NPP gradually strengthened over time. (3) High levels of PM_2.5_ concentration and CO_2_ emissions had negative effects on NPP, while high levels of MAT and MAP had positive effects on NPP.

## 1. Introduction

Net primary productivity (NPP) refers to the organic carbon fixed by plants through photosynthesis [1,2]. Ecosystems with high productivity not only maintain high biodiversity but also play a critical role in conserving water and soil nutrients [3]. However, it is still unclear whether global warming will significantly affect ecosystem productivity, despite the changing distribution patterns and dominant species of vegetation in recent years [4]. Additionally, the greenhouse effect resulting from global warming has an obvious spatial–temporal impact. Thus, exploring the spatiotemporal changes in the relationship between vegetation productivity and global warming can provide a theoretical basis for predicting the impact of global change on forest function [5,6,7].

In recent years, China’s rapid economic development and urbanization have resulted in severe air pollution problems, particularly with regard to haze (atmospheric particulate: PM). Numerous studies have shown that PM_2.5_, particulate matter with a diameter of 2.5 microns or less in the atmosphere, is the main pollutant in many Chinese cities [8]. PM_2.5_ absorbs and scatters sunlight, which alters surface temperature and affects precipitation in various ways [9]. The interaction between climate factors and PM_2.5_ may impact NPP, making the relationship between PM_2.5_ emission and NPP a crucial part of this study.

Previous research has shown that global warming may have an impact on the productivity of terrestrial ecosystems [10]. Since the beginning of the industrial revolution, the continuous increase in fossil fuel emissions has led to a sharp rise in the concentration of pollutants such as CO_2_ and PM_2.5_ in the atmosphere. A significant increase in CO_2_ concentrations will result in global warming [11]. Current research indicates that the Earth may warm by 2–7 °C by the end of this century. Global warming will cause changes in plant photosynthesis, plant respiration, and organic matter decomposition, which may impact NPP [12]. However, the mechanism of NPP changes caused by global warming remains unclear. Given China’s increasing pressure to reduce CO_2_ emissions, investigating the mechanism of NPP fluctuations caused by climate change can help us better mitigate the adverse effects of climate warming in the future and formulate policies to address climate warming.

Numerous studies have found that rising levels of CO_2_ caused by global warming will lead to a decrease in precipitation [13], and temperature and precipitation synergistically affect the accumulation of NPP [14]. PM_2.5_ and CO_2_ are the primary substances that cause global warming. PM_2.5_ changes the net photosynthetic rate by affecting plant photosynthesis, thereby affecting ecosystem NPP [15,16,17,18]. NPP is an essential variable to characterize plant activities and a crucial factor in determining ecosystem carbon sinks and regulating ecological processes [19]. Estimating vegetation NPP on a large spatial and temporal scale using multisource remote sensing data and comprehensive spatial pattern and dynamic analysis with GIS technology has become an important method to quantify NPP [20,21]. Nonetheless, uncertainties remain in the relationship between NPP and climate change to global warming.

This study employed MAT, MAP, PM_2.5_, CO_2_, and NPP data from 1137 sites across China from 2001 to 2017 to investigate (1) the spatiotemporal trends of NPP, MAT, MAP, PM_2.5_ concentration, and CO_2_ emissions; (2) the relationship between NPP and MAT, MAP, PM_2.5_ concentration, and CO_2_ emissions; and (3) the impact of different levels (high/low) of MAT, MAP, PM_2.5_ concentration, and CO_2_ emissions on NPP.

## 2. Results

From 2001 to 2017, MAT and MAP exhibited a significant upward trend (Figure 1A,B). The concentration of PM_2.5_ increased from 2001 to 2011, and then began to decline after 2011 (Figure 1C). CO_2_ emissions consistently increased during this period (Figure 1D). NPP decreased from 2001 to 2011 initially, and then increased from 2012 to 2017, exhibiting an opposite trend to that of PM_2.5_. Over time, NPP displayed a decreasing-then-increasing trend, with its lowest value being in 2011 (Figure 2). As time progressed, NPP and climatic factors showed clear regional differences (Figure 3 and Figure 4). In general, productivity and climate factors exhibited significant habitat heterogeneity.

MAT and MAP were significantly positively correlated with NPP (*R^2^* = 0.23, *p* < 0.001; *R^2^* = 0.24, *p* < 0.001) (Figure 5A,B). However, PM_2.5_ concentrations and CO_2_ emissions were significantly negatively correlated with NPP (*R^2^* = 0.04, *p* < 0.001; *R^2^* = 0.06, *p* < 0.001) (Figure 5C,D).

Although MAT, MAP and NPP has shown a positive correlation over time, the positive correlation is gradually weakening. PM_2.5_ concentrations, CO_2_ emissions and NPP have been negatively correlated. However, the negative correlation is gradually increasing (Figure 6A,B). High levels of MAT and MAP have a positive effect on NPP. However, high levels of PM_2.5_ concentration and CO_2_ emissions have a negative effect on NPP (Figure 7).

## 3. Discussion

We investigated the spatiotemporal changes in forest NPP in China from 2001 to 2017. NPP was calculated based on the CASA model, although its prediction accuracy is not perfect due to the differences in spatial resolution and the actual and simulated values of variable factors [22]. Nevertheless, it is widely recognized and applied in numerous macroecological studies [23,24,25].

Changes in various environmental factors, such as temperature, precipitation, PM_2.5_ concentration, and CO_2_ emissions, caused by global warming can impact the interannual variation in NPP [6,26]. Greenhouse gas emissions have become an urgent ecological problem globally, making it crucial to explore the relationship between various climate factors and NPP to understand the impacts of climate warming on ecosystem functions [27].

Temperature is one of the main climatic factors that affect ecosystem productivity [28,29]. In our study, MAT exhibited an increasing trend from 2001 to 2017. NPP demonstrated a significant positive correlation with temperature, which is consistent with the results of Gu et al. [30]. Rising temperatures can increase the diversity and richness of functional genes of photosynthetic microorganisms in soil, as well as the activity of soil microorganisms [31]. Under rising temperatures, nutrient availability and the utilization efficiency required for forest growth can also improve, promoting soil nutrient mineralization and accelerating nutrient release, which can contribute to the improvement of vegetation NPP [32]. Temperature increase can also accelerate plant photosynthesis, stimulate plant growth, improve carbon absorption, and thus increase vegetation NPP [33,34,35].

Precipitation has also been found to be one of the climatic factors affecting ecosystem productivity [36]. NPP significantly increased with an increase in MAP [12]. The availability of nutrients in soil increases with increasing water content, and the effect of precipitation on soil nutrient availability can affect plant photosynthesis, thus improving vegetation NPP [37,38].

PM_2.5_ has attracted widespread attention in recent times [38]. However, few studies have investigated the influence of PM_2.5_ on vegetation NPP. We found a significant negative correlation between PM_2.5_ concentration and NPP, which may have been due to plants adsorbing particulate matter through leaf holes or adsorbing particulate matter onto the leaf surface. As a result, plants reduce the absorption efficiency of external oxygen, water, and sunlight during photosynthesis, thus reducing the NPP of plants [39].

Since the industrial revolution, the significant increase in the use of fossil fuels has led to a continuous increase in CO_2_ emissions, which is a primary factor contributing to global warming. The increase in CO_2_ emissions significantly affects vegetation NPP [40]. We found a significant negative correlation between CO_2_ emissions and NPP. With an increase in CO_2_ emissions, the carbon sink will fail prematurely, and leaves will age quickly, leading to a decrease in plant NPP [41,42]. The interspecific survival–growth tradeoff theory suggests that, in the case of higher CO_2_ concentrations, faster plant growth would be accompanied by a higher mortality risk, thus affecting plant NPP [43]. The above results are consistent with the findings of numerous studies in recent years [5,44,45]. However, due to the significant physiological spatiotemporal differences of plant individuals, the calculation of NPP by satellite also involves some uncertainty.

The results of this study indicate that high emissions of PM_2.5_ and CO_2_ have serious negative effects on NPP. Therefore, we should pay close attention to the negative effects of increasing PM_2.5_ and CO_2_ emissions on plant NPP. It is crucial to deeply understand the driving mechanism of the terrestrial carbon cycle, increase carbon fixation, reduce carbon emissions, and formulate air pollution prevention measures in a timely way to mitigate the negative impact of climate warming on the ecosystem.

## 4. Study Area and Methods

### 4.1. Data Collection

MAT and MAP data for China from 2000 to 2017 were extracted from the WorldClim global climate database with a spatial resolution of 1 km [45]. PM_2.5_ data with a 1 km spatial resolution were extracted from a published database [46]. CO_2_ data were extracted from a database derived from ODIAC (https://download.csdn.net/download/sooluo/73733136?utm_source=bbsseo (accessed on 20 October 2022) with a spatial resolution of 1 km.

NPP data from 2000 to 2017 were obtained at a 250 × 250 m resolution from NASA (https://search.earthdata.nasa.gov/search) (accessed on 10 October 2022). The Carnegie-Ames-Stanford Approach (CASA) model was used to estimate NPP as follows [22]:(1)NPP(x,t)=APAR(x,t)×ε(x,t)
where *APAR*(*x*, *t*) represents the photosynthetically active radiation (PAR, in units of MJ/m^2^) absorbed at pixel *x* in month *t*, and *ε*(*x*, *t*) represents the actual light energy utilization at pixel *x* in month *t* (g C/MJ).

### 4.2. Data Analysis

To examine the contribution of climatic factors (MAT, MAP, PM_2.5_, and CO_2_ emissions) to the spatial variation in annual NPP, a linear regression model was used, with R^2^ being used to assess the model’s goodness-of-fit. Linear regression was conducted using the R package ‘lme4’ [45].

To evaluate the effects of time (year) on the variables, we used Generalized Additive Models (GAMs), which utilize both parametric and non-parametric components to reduce the model risks inherent in linear models. All calculations were conducted within the R environment using the ‘mgcv’ package [47].

Furthermore, we examined the effects of different concentrations of climate factors on NPP by dividing all the factors into three parts according to the calculated value, with the middle part serving as the reference variable. The effect size of climatic variables was analyzed based on the following model:ln*RR* = ln(X_e_/X_c_)(2)
where X_e_ and X_c_ represent the mean values of the variables.

## Figures and Tables

**Figure 1 plants-12-01404-f001:**
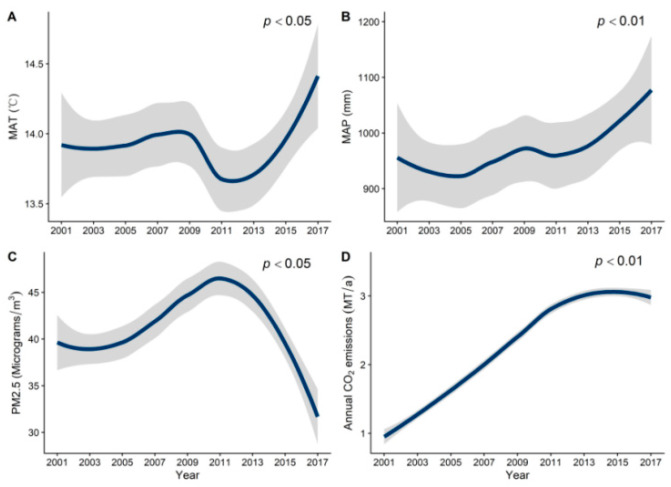
GAMs were used to predict the effects of time on MAT (**A**), MAP (**B**), PM_2.5_ concentration (**C**), and CO_2_ emissions (**D**) in China from 2000–2017, respectively, (*p* < 0.05).

**Figure 2 plants-12-01404-f002:**
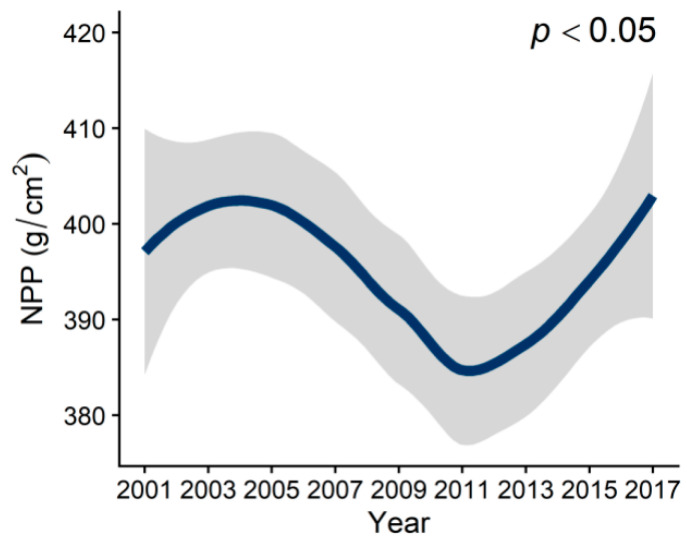
GAMs were used to predict the effects of time on NPP (*p* < 0.05). NPP showed a trend of decreasing first and then of rising over time. In 2011, the value of NPP reached the lowest.

**Figure 3 plants-12-01404-f003:**
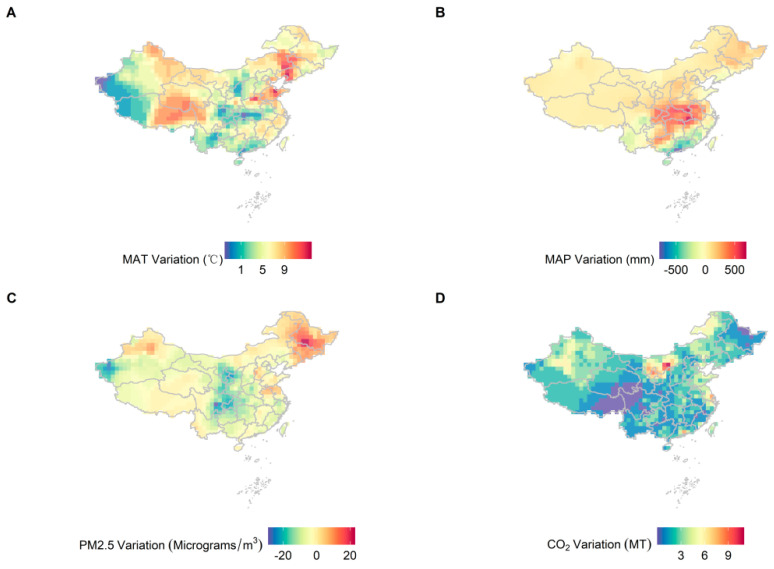
Variations in MAT, MAP, PM_2.5_ concentrations and CO_2_ emissions in China from 2001 to 2017. The spatial resolution is 1 km. (**A**) MAT; (**B**) MAP; (**C**) PM_2.5_ concentration; (**D**) CO_2_ emissions.

**Figure 4 plants-12-01404-f004:**
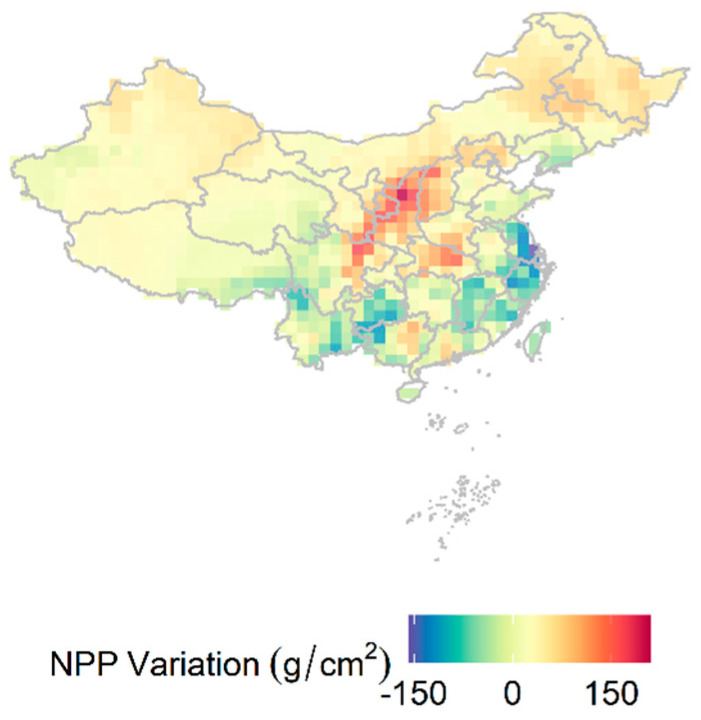
Variations in NPP in China from 2001 to 2017. The spatial resolution is 1 km. On the whole, there is a large number of regions in southern China where NPP is declining gradually.

**Figure 5 plants-12-01404-f005:**
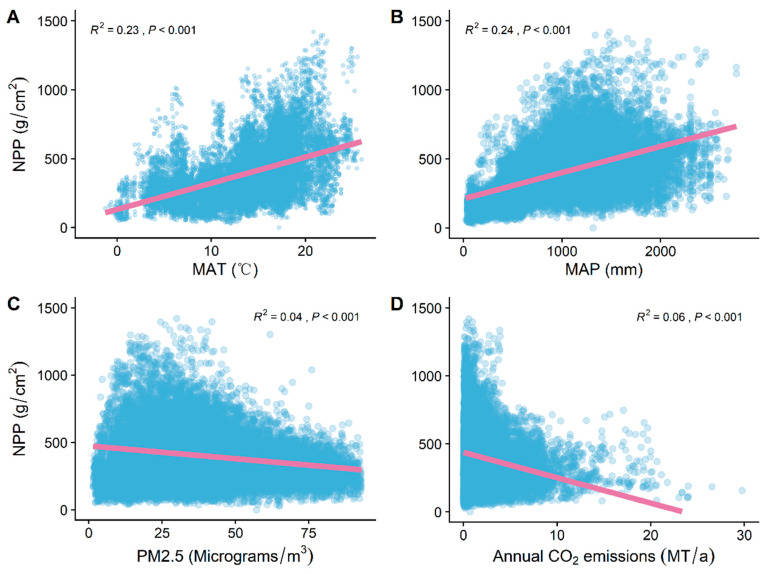
General linear correlation analysis of (**A**) MAT, (**B**) MAP, (**C**) PM_2.5_ concentrations and (**D**) CO_2_ emissions with NPP. *R^2^* represents how well the model fits the variables studied, and the *p* value represents the significance level.

**Figure 6 plants-12-01404-f006:**
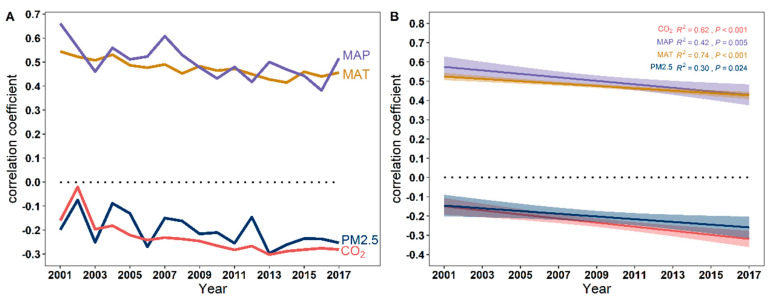
(**A**) The correlation between environmental factors (CO_2_ emission, MAP, MAT, and PM_2.5_ concentration) and NPP over time from 2001 to 2017. (**B**) The plots of the GAMs’ smooth function for indicating the effects of time (year) on the correlation coefficient.

**Figure 7 plants-12-01404-f007:**
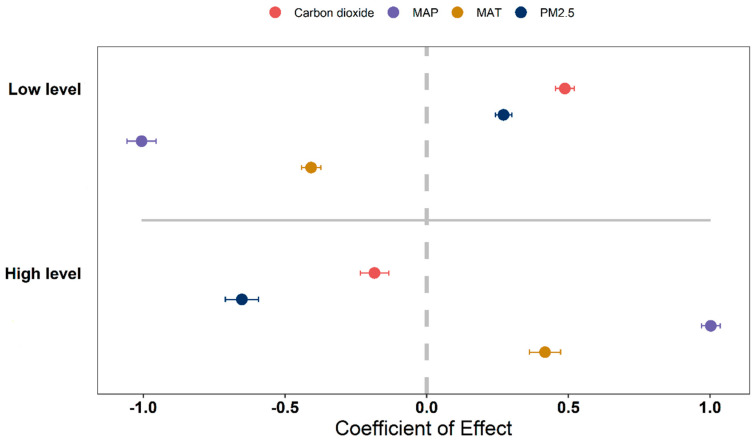
The effects of different environmental factors on high and low levels on NPP were analyzed, and the mean confidence interval of ±95% of the effect size is reported. The dotted line indicates that the effect size is zero. The environmental factors considered were CO_2_ emissions, MAP, MAT, and PM_2.5_ concentration, and were classified into three levels: high, medium, and low. The high and low levels were the experimental groups, while the medium level was the control group. For MAT, the low level was less than or equal to 10 °C, the medium level was greater than 10 °C but less than 20 °C, and the high level was greater than or equal to 20 °C. For MAP, the low level was less than or equal to 400 mm, the medium level was greater than 400 mm but less than 800 mm, and the high level was greater than or equal to 800 mm. For PM_2.5_ concentration, the low level was less than or equal to 35 μg/m^3^, the medium level was greater than 35 μg/m^3^ but less than 75 μg/m^3^, and the high level was greater than or equal to 75 μg/m^3^. For CO_2_ emissions, the low level was less than or equal to 2 MT/a, the medium level was higher than 2 MT/a but less than 5 MT/a, and the high level was greater than or equal to 5 MT/a.

## Data Availability

Not applicable.

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
