# Peer review of "Time Effects of Global Change on Forest Productivity in China from 2001 to 2017"

_plants, 2023, doi:10.3390/plants12061404_

Round 1
Reviewer 1 Report (Previous Reviewer 2)
Since the authors have not heeded my comments from the first round /first submission, therefore I do insist that they do so now.
2) I insist that some info, however brief, about the methodology of obtaining the data for analyses, should by all means be added in the abstract; It is in the interests of authors themselves, as well as for the Journal.
2) I also once again strongly recommend the authors to address, however briefly, the uncertainties of the satellite methodology to estimate NPP either in Introduction or Discussion, or better both. Introduction shall also benefit from adding some detailed text and references about the methods to estimate NPP, otherise it is incomplete.
Author Response
Dear Editors,
We really greatly appreciate you for processing our manuscript entitled “Time Effects of Global Change on forest Productivity in China from 2001 to 2017” (Manuscript ID: plants-2268726). We are grateful for your and the reviewers’ valuable suggestions and comments on the manuscript. We have polished the whole manuscript based on the comments, in order to ensure that it is clear and as brief as possible, following the Plants format.
The point-by-point responses to your and the reviewers’ comments can be found below. The comments are shown in black font and the responses are shown in blue font.
Yours sincerely,
Jie Gao and coauthors
########### Point-to-Point Response Letter ###############
Reviewer #1 (Remarks to the Author):
- I insist that some info, however brief, about the methodology of obtaining the data foranalyses, should by all means be added in the abstract; Itis in the interests of authors themselves, as well as for the Journal.
Response: Thank you very much for your advice. We revised the Abstract according to your opinion. In the resubmitted paper, we added the method to obtain NPP data based on CASA (Carnegie-Ames-Stanford Approach) model of remote sensing satellite in the abstract.
- I also once again strongly recommend the authors to address, however briefly, the uncertainties of the satellite methodology to estimate NPP either in introduction or Discussion, or better both. Introduction shall also benefit from adding some detailed text and references about the methods to estimate NPP, otherise it is incomplete.
Response: Thank you very much for your advice. We have added to the discussion section of the resubmitted article a discussion of the inaccuracy of satellite methods for estimating NPP. NPP was calculated based on the CASA model, although its prediction accuracy is not perfect due to the differences in spatial resolution and the actual and simulated values of variable factors. Nevertheless, it is widely recognized and applied in numerous macroecological studies.
At the same time, we have added detailed text of the satellite methodology to estimate NPP in the introduction. Estimating vegetation NPP at a large spatial and temporal scale using multi-source remote sensing data and comprehensive spatial pattern and dynamic analysis with GIS technology has become an important method to quantify NPP. The methods for estimating NPP are also supplemented with corresponding references.
Reviewer 2 Report (New Reviewer)
We have a number of comments on this manuscript.
1. In the Introduction, we ask the authors to decipher in more detail what PM 25 is? (not all readers can know much about this term)
2. It is not entirely clear what global conclusion can be drawn from the results of this manuscript? How to deal with climate warming? What should be the policy of the peoples of the entire planet in this area?
The results represent data for China only. To what extent can they be transferred to the entire planet?
The answers to these questions should be included in the Discussion.
Author Response
Dear Editors,
We really greatly appreciate you for processing our manuscript entitled “Time Effects of Global Change on forest Productivity in China from 2001 to 2017” (Manuscript ID: plants-2268726). We are grateful for your and the reviewers’ valuable suggestions and comments on the manuscript. We have polished the whole manuscript based on the comments, in order to ensure that it is clear and as brief as possible, following the Plants format.
The point-by-point responses to your and the reviewers’ comments can be found below. The comments are shown in black font and the responses are shown in blue font.
Yours sincerely,
Jie Gao and coauthors
##### Point-to-Point Response Letter ###############
Reviewer #2 (Remarks to the Author):
- In the introduction, we ask the authors to decipher in more detail what PM 25 is? (not all readers can know much about this term)
Response: Thank you very much for your advice. We have added the explanation of PM2.5 in the introduction according to your comments. PM2.5 is particulate matter with a diameter of 2.5 microns or less in the atmosphere.
2.It is not entirely clear what global conclusion can be drawn from the results of this manuscript? How to deal with climate warming? What should be the policy of the peoples of the entire planet in this area?
The results represent data for China only. To what extent can they be transferred to the entire planet?
Response: Thank you very much for your advice. There are some errors of expression in the article. We have changed the global change in the article to climate change in China. And we added measures to deal with climate change in the discussion part of the article. We should pay close attention to the negative effects of increasing PM2.5 and CO2 emissions on plant NPP. It is crucial to deeply understand the driving mechanism of the terrestrial carbon cycle, increase carbon fixation, reduce carbon emissions, and timely formulate air pollution prevention measures to mitigate the negative impact of climate warming on the ecosystem.
This manuscript is a resubmission of an earlier submission. The following is a list of the peer review reports and author responses from that submission.
Round 1
Reviewer 1 Report
Plants, Jan. 2023
Article: Time effects of Global Change of forest …
Authors: Wang J. et al.
Comments for the authors
This manuscript offers many not understandable results and causes a bunch of problems and questions:
1. It seems that the basic data are given for the whole country of China. Is the whole country covered with forests?
2. Why do the authors focus on very small increases of fine particles and [CO2] from 2001-2017? It is known that fine particles reduce net photosynthesis by clogging the stomata, and on the other hand, increasing [CO2] causes higher net photosynthesis rates. Why do the authors not consider this profound antagonism? By the way, it is known that a doubling of the preindustrial atmospheric [CO2] (~270 → 540 ppm) leads to an increase of net photosynthesis of ~30% on average in juvenile tree stands (with large deviations). Between 2001 and 2017 only an increase of a few ppm CO2 and of a few more or constant fine particles (very local) occurred. This reviewer is convinced that any effect of such small alterations cannot really be detected on that scale of a huge country with a lot of different land use forms.
3. Why do the authors not consider other much more important factors affecting NPP (net primary production) like desertification, gaseous pollutants, heavy metals, loss of forested areas between 2001 and 2017 (especially near quickly growing cities), overuse of forested areas, drainage of wetlands, soil sealing, soil degradation etc. All these factors influence NPP quite a lot more on a country scale than [CO2] and fine particles.
4. Another point is that changes of temperature and precipitation are certainly quite different and have different effects in the South of China as in the North-West and North of China. Why was this not be considered exactly?
Summary
Unfortunately, this reviewer cannot support the publication
Reviewer 2 Report
I read the manuscript with great interest, and believe that it will be interesting for readers when published. However, for the purpose of increasing the clarity and comprehensiveness, I would strongly recommend the authors
1) to squeeze some info about obtaining the data for analyses in the abstract;
2) to address, however briefly, the uncertainties of the satellite methodology to estimate NPP either in Introduction or Discussion, or better both.
In addition, I would like to indicate that it is not a proper way to begin a sentence with an abbreviation, and recommend to edit such.